# Hyaluronidase 6 Does Not Affect Cumulus–Oocyte Complex Dispersal and Male Mice Fertility

**DOI:** 10.3390/genes13050753

**Published:** 2022-04-25

**Authors:** Hyewon Bang, Sujin Lee, Pil-Soo Jeong, Dong-Won Seol, Daeun Son, Young-Hyun Kim, Bong-Seok Song, Bo-Woong Sim, Soojin Park, Dong-Mok Lee, Gabbine Wee, Joon-Suk Park, Sun-Uk Kim, Ekyune Kim

**Affiliations:** 1College of Pharmacy, Catholic University of Daegu, Gyeongsan-si 38430, Korea; sw82100@naver.com (H.B.); sujinlee@cu.ac.kr (S.L.); agsilver119@gmail.com (D.S.); 2Futuristic Animal Resource and Research Center, Korea Research Institute of Bioscience and Biotechnology, Cheongju-si 28116, Korea; spectrum@kribb.re.kr (P.-S.J.); sunuk@kribb.re.kr (S.-U.K.); 3Preclinical Research Center, Daegu-Gyeongbuk Medical Innovation Foundation (DGMIF), Daegu 41061, Korea; seol@kmedihub.re.kr (D.-W.S.); gabbine@kmedihub.re.kr (G.W.); jsp@kmedihub.re.kr (J.-S.P.); 4National Primate Research Center (NPCR), Korea Research Institute of Bioscience and Biotechnology, Cheongju-si 28116, Korea; kyhq@kribb.re.kr (Y.-H.K.); sbs6401@kribb.re.kr (B.-S.S.); embryont@kribb.re.kr (B.-W.S.); 5Department of Biochemistry and Molecular Biology, Melvin & Bren Simon Comprehensive Cancer Center, Indiana University School of Medicine 635 Barnhill Drive, Indianapolis, IN 46202, USA; parksj678@gmail.com; 6Biomedical Manufacturing Technology Center, Korea Institute of Industrial Technology, Yeongcheon-si 38822, Korea; cowboyle@kitech.re.kr; 7Biohealth Convergence Research center for East sea Rim, Catholic University of Daegu, Gyeongsan-si 38430, Korea

**Keywords:** hyaluronidase, HYAL6, sperm, fertility, mouse model, COC dispersal

## Abstract

Glycosylphosphatidylinositol-anchored sperm hyaluronidases (HYAL) assist sperm penetration through the cumulus–oocyte complex (COC), but their role in mammalian fertilization remains unclear. Previously, we demonstrated that sperm from HYAL 5 and 7 double-knockout (dKO) mice produced significantly less offspring than sperm from wild-type mice due to defective COC dispersal. However, the HYAL6 gene remained active in the sperm from the dKO mice, indicating that they were not entirely infertile. This study explored the role of HYAL6 in fertilization by analyzing HYAL6-mutant mice. In this mouse model, HYAL5 and HYAL7 were present in the HYAL6-knockout sperm, and they could disperse hyaluronic acid. We found that HYAL6 was present on the surface of sperm. However, male mice lacking the HYAL6 gene had normal fertility, testicular integrity, and sperm characteristics. Furthermore, in vitro fertilization assays demonstrated that HYAL6-deficient epididymal sperm functioned normally. Therefore, HYAL6 is dispensable for fertilization.

## 1. Introduction

The data obtained from the human and mouse genome projects indicate that mammals have approximately 30,000 genes. Functional analyses are performed to evaluate the physiological functions of genes, and gene knockout (KO) systems facilitate these analyses. In the late 1980s, KO mouse models were developed once engineering sciences disciplines were established [1,2]. Since 2010, the clustered regularly interspaced short palindromic repeats (CRISPR)/CRISPR-associated protein 9 (Cas9) gene-editing tool has been used to generate KO mice models within a shorter production time than before [3]. Gene malfunctions related to homeostasis generally cause cell death. However, some sperm-specific genes are not critical for cell survival [4,5,6]. Typically, genes manipulated during fertilization are not associated with life-threatening diseases; however, they sever the flow of genetic information to the offspring, thereby contributing to species extinction. Fertilization is the critical process of conceiving new offspring [7]. During this process, ejaculated sperm approach the cumulus–oocyte complex (COC) formed in the ampulla of the uterine tube, one of the female reproductive organs of mammals [7,8]. However, aberrant spermatogenesis is associated with male sterility because it prevents the sperm from approaching the oocyte.

Hyaluronidases (HYAL) are extracellular membrane proteins found in sperm and expressed in somatic (e.g., HYAL1, HYAL2, and HYAL3) and germ (e.g., HYAL5, HYAL6, and HYAL7) cells [9]. Because HYAL7 was identified from the posterior head of the spermatozoon, it was initially called PH-20 but was later renamed sperm adhesion molecule 1 (i.e., SPAM1) [10,11]. In 2002, HYAL7 functional assays performed using KO mice revealed that male HYAL7 KO mice had normal fertility [12]. Interestingly, in 2005, a novel HYAL was identified in the extracellular matrix of sperm in male HYAL7 KO mice. It was expressed in a pattern different from HYAL7 and was called HYAL5 [13]. HYAL5 KO mice also had normal fertility [14].

Although HYAL5 and HYAL7 are not independently essential for fertilization, HY-AL5 and HYAL7 double-KO (dKO) mice were significantly but incompletely infertile [15]. It is not clear why complete infertility did not occur. One hypothesis is that HYAL6 participates in fertilization and increases the fertilization rate. It has been found that mouse HYAL6 is composed of 500 amino acids and possesses HYAL- and zona pellucida (ZP)-binding domains. However, no research has been conducted on the function of HYAL6 in mammalian fertilization. Therefore, we generated a HYAL6 KO mouse model using CRISPR/Cas9 to determine the role of HYAL6 during fertilization in mammals.

## 2. Materials and Methods

### 2.1. Total RNA Extraction and Reverse Transcription (RT)-Polymerase Chain Reaction (PCR)

Total RNA was obtained from HYAL6^+/+^ and HYAL6^−/^^−^ mouse testicular cells using Isogen (Takara, Shiga, Japan), as previously described. Total RNA (5 µg) was reverse-transcribed to complementary DNA (cDNA) using the SuperScript III First-Strand Synthesis System (Thermo Fisher Scientific, Waltham, MA, USA). PCR amplification was performed using Taq DNA polymerase (ELPIS-BIOTECH, Daejeon, Korea) following the manufacturer’s instructions. One-tenth of the first strand of the cDNA reaction mixture was used as the template. Specific PCR primers were designed to amplify the mouse genes *HYAL2*, *HYAL5*, *HYAL6*, *HYAL7*, and glyceraldehyde-3-phosphate dehydrogenase (i.e., *GAPDH*). The oligonucleotide sequences of all primers used in this study were as follows: HYAL2: 5′-AGACCCCAGCACCTGTGGGGCT-3′ (forward), 5′-TACTGGGTGGCCCAGGACACGT-3′ (reverse), HYAL5: 5′-TTTCACAGAAGCTGTCAAGTGG-3′ (forward), 5′-G TAATCATGTAAGTCCAGA-CAA-3′ (reverse), HYAL6: 5′-TAGGAATTGGGGCAGTAGAATG-3′ (forward), 5′-ACTTTCGCCAATTGTGTGCACT-3′ (reverse), HYAL7: 5′-AAGGAAGTTTATGGAAGGAACT-3′ (forward), 5′-GAGCAACAATTTCACCAATTGT-3′ (reverse), GAPDH: 5′-AGATTGTCAGCAATCATCCTG-3′ (forward), and 5′-TGCTTCACCACCTTCTTGATGT-3′ (reverse). The PCR protocol was 35 cycles of 94 °C for 60 s, 60 °C for 60 s, and 72 °C for 60 s. The amplified cDNA products were analyzed using 1.5% agarose gel electrophoresis, as previously described.

### 2.2. Animals

The wild-type (WT) mice (C57BL/6J) were provided by Hyochang Science Co. Ltd. (Daegu, Korea), and Dr. Sim’s group generated the HYAL6 KO mice at KRIBB. All animals used in the experiment were housed at a constant temperature of 23 ± 1 °C and humidity of 50 ± 20%, with free access to food and water. All animal experiments performed in this study were approved by the Institutional Animal Care and Use Committee of the Daegu-Gyeongbuk Medical Innovation Foundation (DGMIF021021602-00).

### 2.3. Establishing and Characterizing the HYAL6 KO Mouse Line

Exon four of HYAL6 was selected as the CRISPR/Cas9 target. Mouse *HYAL6* consists of four exons on the sixth chromosome. Guide RNAs (gRNAs) were designed, synthesized, and ligated into a pX330 plasmid to target the DNA sequence of exon four in the *HYAL6* gene, which makes up HYAL domain. Before injecting the constructs into the mouse embryos, the gRNAs were validated in the NIH3T3 cells to ensure that they could target and cut the desired HYAL6 regions. After validation, the plasmid containing the gRNAs and Cas9 were microinjected into mouse two-cell embryos and transferred into the pseudo-pregnant females to target exon four in the *HYAL6* gene. To confirm the *HYAL6* gene mutation, pairs of forward and reverse primers (P1: 5′-ATGTTTATCCAGTGGGTGACACAG-3′ and P2: 5′-CCAGTCTATAACAGCAAGTCCTTC-3′) were designed to amplify the CRISPR-targeted region. Successful amplification using this primer pair should indicate if the allele was from WT or KO mice. However, the size difference between the two bands is minimal. Therefore, we performed gene sequencing. 

### 2.4. Preparation of Agarose Gels and Electrophoresis of PCR Products

To confirm the CRISPR induced 17-bp deletion in *HYAL6*, approximately 370-bp PCR products from F1 pups were gel extracted and sub-cloned into pGEMT vectors (A3600, Promega, Madison, WI, UAS) by TA cloning and analyzed by Sanger sequencing. To be more specific, a 4% agarose gel was prepared using low EEO agarose (A5093) of a 95% purity (Sigma–Aldrich, Saint Louis, MO, USA). It was dissolved in a 1× Tris-Acetate EDTA (TAE) buffer (40 mM Tris-acetate, 1 mM EDTA, pH 8.0) by heating the solution in a microwave oven for 3 min, after which it was immediately poured onto a gel casting tray fitted with a 7-well comb. Next, 25 µL of each PCR sample was loaded into each well and electrophoresis was performed for 2 h at 6.7 volts/cm (Optima Inc., Tokyo, Japan). The TAE buffer in the electrophoresis was changed repeatedly during the operation. After 10 mins of electrophoresis, gel bands were observed reacting to approximately 1% of a EtBr (ethidium bromide) solution. The bands were made visible with ethidium bromide, and then excised from the gels using a razor blade. After the fragments were soaked in water briefly to remove the gel buffer and ethidium bromide, DNA purification was performed as described by Qiagen [6].

### 2.5. Protein Extract Preparation

Epididymal sperm were prepared from WT and KO mice, as previously described. The sperm were suspended in a lysis buffer (pH 7.4) consisting of 20 mM tris-hydrochloric acid, 1% Triton X-100, 150 mM sodium chloride (NaCl), and a 10% protease inhibitor cocktail. It was then kept on ice for 20 min. The protein concentrations were determined using the Bradford protein assay.

### 2.6. Antibody Preparation

An anti-HYAL6 antibody was prepared by immunizing female rabbits with poly-peptides, produced by Cosmo Genetech (Seoul, Korea), containing part of the mouse HYAL6 domain corresponding to residues 203–228. The samples were coupled to keyhole limpet hemocyanin, injected into the muscle three times in two-week intervals, and the serum was collected to confirm the mouse anti-HYAL6 antibody. The antisera were fractionated using ammonium sulfate (33% saturation), and the mouse anti-HYAL6 antibody was then affinity-purified on a Sepharose 4B column conjugated with bovine serum albumin-bound HYAL6 domain polypeptides, as described previously. Finally, dialysis was performed on the purified anti-HYAL6 antibody three times using 1× phosphate-buffered saline (PBS).

### 2.7. Zymography Assay

The HYAL activity of sperm extracts was made visible by sodium dodecyl sulfate (SDS)-polyacrylamide gel electrophoresis (10%) with a 0.05% human umbilical cord hyaluronan under non-reducing conditions. The gels were washed with a 50 mM sodium acetate buffer (pH 6.0) containing 0.15 M NaCl and 3% Triton X-100 at room temperature for 2 h to remove the SDS and then incubated in the same buffer without Triton X-100 at 37 °C overnight. The hyaluronan-hydrolyzing proteins were detected as transparent bands against a blue background by staining the gels with 0.5% Alcian Blue 8 GX and Coomassie brilliant blue R250 [15].

### 2.8. Histological Analysis

The testes of five-month-old HYAL6^−/^^−^ mice were fixed in PBS containing 4% paraformaldehyde, washed with PBS, and then frozen in an optimal cutting temperature compound (Gibco, Billings, MT, USA). The sections were prepared using a Leica microtome SM2000R (Wetzlar, Germany), stained with hematoxylin and eosin (H&E) as previously described, and then made visible using a Leica microscope [6].

### 2.9. Fertility Testing

Sexually mature male HYAL6^−/^^−^ and WT mice were bred with WT female mice over two months old from a C57BL/6N × DBA/2cross (also called B6D2F1 mice) for five months. The pups were counted and separated from the mother within one week after birth to allow the mother to prepare for the subsequent breeding. The presence of vaginal plugs confirmed copulation, which was checked every morning.

### 2.10. Epididymal Sperm Analyses with Computer-Assisted Sperm Analysis (CASA) Systems

Sperm motility and motion kinetics were measured using the CASA system. Epididymal regions from eight-month-old WT and HYAL6 KO mice were dissected. The cauda was carefully trimmed to remove adipose and other tissues, rinsed in 1× PBS, and placed in 300 μL of a human tubal fluid (HTF) medium (Cosmo Bio, Tokyo, Japan). Multiple incisions were made in the cauda to release sperm into the media; incubating for 10 min at 37 °C and 5% carbon dioxide (CO_2_) prompted their release. After the incubation, the tissue was removed, and the suspension was gently mixed by swirling. The suspension was diluted at 1:20 in an HTF complete medium to approximately 1.45 × 10^6^ sperm/mL. The sperm were analyzed for motility and concentration using CASA by loading 10 µL of the sperm suspension into a 20 μm deep Leja slide chamber. The percentages of the total, the progressively, and the rapidly motile sperm were recorded for each sample, as were the average path, the curvilinear path, the straight-line velocities, and the amplitude of lateral head displacement (μm). Sperm with ≥70% straightness and a velocity average path (VAP) of ≥50 μm/s were classified as progressively motile, and sperm with a VAP of ≥50 μm/s were classified as rapidly motile.

### 2.11. COC Dispersal In Vitro

We collected eggs surrounded by a dense layer of cumulus cells from the oviductal ampulla of the super-ovulated mice 16 h after human chorionic gonadotropin injection, placing them in a 0.2 mL drop of a Toyoda, Yokoyama, Hoshi (TYH) medium [16] covered with mineral oil. Fresh cauda epididymal sperm from three-month-old dKO mice were capacitated by incubating them in a 0.2 mL drop of TYH medium for 2 h at 37 °C and 5% CO_2_. 

The KO sperm or sperm extracts (4 μg protein/μL) were added to the eggs in a TYH medium and incubated for 1 to 3 h at 37 °C and 5% CO_2_. We used a Nikon ECLIPSE Ti-S (Tokyo, Japan) microscope equipped with a Nikon D5-Qi1Mc digital camera to evaluate the COC morphology. The sperm motility assays were conducted in a TYH medium with 0.4% hyaluronic acid. Briefly, dKO sperm were incubated in the medium at 37 °C for 30 min and observed using the Nikon microscope.

### 2.12. In Vitro Fertilization (IVF) Assay

We induced super-ovulation of six-week-old C57BL/6N mice using intraperitoneal injections of 7.5 IU pregnant mare serum gonadotropin and human chorionic gonadotropin at 46 h intervals. The COC was collected from the ampulla of the oviduct, and denuded oocytes were prepared by treating the COC with 0.1% HYAL. The oocytes were cultured in an M16 medium before insemination. Cauda epididymides were dissected from WT and KO mice, gently squeezing to collect the spermatozoa in a modified HTF (mHTF) medium. The spermatozoa were capacitated at 37 °C for 1 h before insemination and mixed with the oocytes at a final concentration of 1.5 × 10^6^ sperm/mL. The oocytes and spermatozoa were co-incubated for 4 h at 37 °C and 5% CO_2_. Next, cumulus-intact or cumulus-free eggs were inseminated by the capacitated sperm (1.45 × 10^5^ sperm/mL) in a 0.2-mL drop of TYH medium. The inseminated eggs were incubated for 6 h at 37 °C and 5% CO_2_. Next, the cumulus cells were removed by incubating the eggs with bovine HYAL (3 units/mL) for 15 min and were washed. The female and male pronuclei of the inseminated eggs were stained with 4′-6-diamidino-2-phenylindole (10 µg/mL) for 30 min and then viewed under a Leica fluorescence microscope.

### 2.13. Statistical Analysis 

The data are presented as means ± standard deviations of at least three independent experiments. After examining the data distributions, we used a Student’s *t*-test in Excel (version X, Microsoft Corporation, Redmond, WA, USA) for statistical comparisons.

## 3. Results

### 3.1. HYAL6 Gene Expression Patterns

Somatic cells in mice express seven *HYAL* genes: *HYAL1*, 2, and 3 on Chromosome 9 and *HYAL4*, 5, 6, and 7 on Chromosome 6 (Figure 1A and Figure 2A). To confirm the expression pattern of the *HYAL6* gene, RT-PCR was performed using cDNA derived from each tissue in the mouse, and it was confirmed that the *HYAL6* gene was specifically expressed in the sperm. *HYAL2*, a representative somatic cell *HYAL* gene, was expressed in all tissues, and *HYAL5* and *HYAL7* were expressed testis, such as *HYAL6* (Figure 1B).

### 3.2. HYAL6 KO Male Mice Have Normal Sperm Parameters

Despite an abundant expression of HYAL6 in mouse sperm, its role in fertilization remains unknown. To determine whether the absence of HYAL6 influences fertility, we produced mutant mice lacking HYAL6 by CRISPR/Cas9 system. Genomic PCR analysis indicated the deletion of 17 nucleotides in the second exon in the HYAL6^−/^^−^ mice (Figure 2B,C and Figure 3A). In addition, an RT-PCR analysis of total cellular RNA from the KO testes demonstrated that HYAL6 completely absence albeit testicular specific *Hyal5*, and *Hyal7* were expressed at levels similar to the levels in WT mice. In addition, an RT-PCR analysis of total cellular RNA from the KO testes demonstrated that HYAL6 was completely absent, while testicular specific HYAL5 and HYAL7 were expressed at levels similar to the levels in WT mice (Figure 3B). Moreover, zymographic assays for Hyal enzyme activity confirmed no significant difference in wild-type and HYAL6 KO sperm extracts (Figure 3C). Figure 3C shows clearly that sperm HYAL activity was completely abolished in the HYAL5/HYAL7 sperm extracts. 

Next, we bred HYAL6 KO and WT males with fertility-proven WT females at least six times. The copulation plugs in mated females formed normally, indicating that HYAL6 KO did not impair the fertilization ability of sperm. Contrary to our expectations, HYAL6 KO breeding pairs produced comparable litter sizes to WT breeding pairs, indicating complete fertility in the KO male mice (Figure 4A). Sperm from six-week-old KO and WT male mice were analyzed using the CASA system to investigate whether the absence of HYAL6 affects sperm function. Similarly, cauda epididymal sperm isolated from KO mice were indistinguishable from that isolated from WT mice regarding shape, motility, and the percentage of acrosome-reacted sperm. In addition, the average number of sperm in the cauda epididymis (1.35 ± 1.31 × 10^7^ sperm/mL; five mice) and the sperm count in the uterus (1.25 ± 1.24 × 10^6^ sperm/mL; five mice) from the KO mice were comparable to that of the WT mice (cauda epididymis: 1.17 ± 1.65 × 10^7^ sperm/mL; uterus: 1.37 ± 1.25 × 10^6^ sperm/mL). These findings indicate no apparent abnormalities in spermatogenesis, sperm maturation, or ejaculation in male KO mice. The appearance and weights of WT and KO mice testes were also comparable and consistent with that of a fertile phenotype (Figure 4B).

### 3.3. Immunostaining of HYAL6

To determine whether the absence of HYAL6 affects fertility, we bred the HYAL6 KO males with fertility-proven WT females and KO females at least five times. Contrary to our expectations, HYAL6 KO breeding pairs produced litter sizes comparable to WT breeding pairs that indicated the complete fertility of KO male mice. The normal formation of copulation plugs in the mated females showed that the fertilization ability of the sperm from KO mice was not impaired. Moreover, the litter sizes of WT and KO mice were normal (average 9.7, and 9.1 offspring, respectively) from crosses between the male and female mice, respectively (Figure 4A). Furthermore, the morphological examinations of H&E-stained spermatogenic cells showed that the cells in the seminiferous tubules in the testes of KO mice had a normal appearance (Figure 4C). Thus, based on these findings, we concluded that HYAL6 is not essential for mouse spermatogenesis. In addition, our affinity-purified anti-HYAL6 antibody gave an immunoreactive signal for the sperm. Expectedly, the immunoreactive signal for HYAL6 disappeared from the HYAL6 KO sperm (Figure 4B).

### 3.4. COC Dispersal and IVF

Adding KO and WT sperm to the COC caused the cumulus cell to easily disperse within the COC, and complete removal occurred three hours after adding sperm (Figure 5A). COC treated with the HYAL5/HYAL7-deficient sperm remained densely packed during the entire three-hour incubation period, similar to that observed in the control COCs incubated without sperm. These results indicate that defective COC dispersion caused by an HYAL-deficient sperm is not related to HYAL6. Moreover, even when the COC dispersal experiment was performed using each WT, HYAL6, and HYAL5/HYAL7-defective sperm extract, the HYAL6 KO sperm extract could disperse the COC, whereas the HYAL5/HYAL7-defective extract could not (Figure 5B). As a result, the deficiency is due to lacking HYAL5 and HYAL7 hyaluronidase activity, which hydrolyzes hyaluronic acids in the extracellular matrix of cumulus cells.

### 3.5. HYAL6 KO Male Mice Exhibit Normal Fertility in IVF

Hyaluronan zymography indicated that two hyaluronan-hydrolyzing proteins with sizes of 55 and 52 kDa are present in the extracts of wild-type sperm [11]. The 55- and 52-kDa hyaluronan-hydrolyzing proteins probably correspond to HYAL5 and SPAM7, respectively, because of no difference between WT and *KO* sperm extracts (Figure 6A). Additionally, Hyal enzyme activity confirmed the complete absence of HYAL activity in the dKO sperm (Figure 6A). To assess the HYAL6 KO mouse sperm and egg interaction, an IVF assay was performed using the capacitated cauda epididymal sperm. When cumulus-intact eggs were used, the fertilization rate after insemination with HYAL6 KO mouse sperm was normal (Figure 6B). No significant difference was found either in the sperm binding to zonapellucida (ZP) or in the fusion of sperm with ZP-free eggs between WT and KO mice (Figure 6C).

## 4. Discussion

The importance of sperm HYAL in mammalian fertilization is well established. Contrary to expectations, a germ cell-specific sperm HYAL gene has not been discovered to date. Therefore, we knocked out sperm-specific HYAL genes one by one to examine their independent roles in mammalian fertilization. We aimed to determine the pattern of HY-AL6 expression and the role of HYAL6 in the fertilization, as the evidence suggested that HYAL6, like HYAL5 and HYAL7, are expressed in a testis-specific manner to participate in COC dispersion during fertilization.

First, we characterized HYAL6 in mice. Using data from the National Center for Bio-technology Information, we found that the amino acid homology of HYAL6, when compared to that of HYAL5 and HYAL7, is 33% and 35%, respectively, and shares a common sequence with basic testis-specific HYAL. Interestingly, some species contain stop codons in the *HYAL6*-gene coding region. Specifically, HYAL6 (HYALP1 in humans) is a shadow gene in human beings but is intact in rodents. The sperm-specific HYAL evolves in various patterns. For example, the sperm-specific *HYAL* genes *HYAL5*, *HYAL6*, and *HYAL7* are expressed in mice, but *HYAL5* does not exist in human beings, and *HYAL6* remained a shadow gene. Therefore, the specific expression patterns remain unclear. Although *HYAL7* is evolutionarily conserved, other genes are removed from the chromosome or become shadow genes; the reason for this remains unknown.

Representative proteins involved in sperm and egg interactions are sperm-specific A disintegrin and metalloprotease (ADAM) molecules [17]. To date, 30 types of ADAM molecules have been discovered, and about half are specifically expressed in germ cells [18,19]. The most well-studied ADAM molecules are fertilin and cyritestin. ADAM1 and ADAM2 comprise the fertilin complex [20,21,22]. Interestingly, ADAM2 is highly conserved in mice and human beings, but ADAM1 is a shadow gene in human beings. ADAM3 is important for the migration of ejaculated sperm through the uterotubal junction and ZP binding in mice but is a shadow gene in human beings [23,24,25]. While the fertilization process in mammals is similar mechanistically, different genes regulate fertilization in mice and human beings.

It is natural that HYAL6 is present in sperm, considering its role in fertilization. Consequently, we succeeded in producing a HYAL6-specific polyclonal antibody. Thus, the presence of HYAL6 in the mouse sperm head was confirmed and its involvement in mammalian fertilization was indicated. Our previous results demonstrated that HYAL5 and HYAL7 showed hyaluronidase activity and played an important role in COC dispersion, but their ZP-binding domains were not directly involved in binding with ZP. Therefore, our final goal was to determine whether HYAL6 functions as a HYAL and affects the dispersal of COC.

We succeeded in creating a HYAL6 KO mouse model using CRISPR/Cas9 on male mice. Contrary to expectations, HYAL6 KO male mice had normal fertility. In addition, reproduction by natural mating proceeded as expected, and the COC dispersal ability and the IVF rate did not differ between the KO and WT mice. We also observed HYAL activity in HYAL6 KO sperm but not in HYAL5/HYAL7 dKO sperm. Therefore, the HYAL activity in the sperm was derived from HYAL5 and HYAL7.

Mouse chromosome 6 contains the reproductive cell-specific *HYAL* genes *HYAL5*, *HYAL6*, and *HYAL7*. As can be seen from Figure 6, the WT mouse sperm shows activity involving two hyaluronidases. When HYAL7 was knocked out of the Baba Group in 2002, there was no 52-kDa band, and when HYAL5 was deleted in 2009, there was no 55-kDa band. When HYAL5 and HYAL7 were deleted at the same time in 2019, it was confirmed that both bands disappeared. HYAL6 KO sperm not only exhibits COC dispersal but also exhibits no difference in hyaluronidase activity from WT (Figure 5 and Figure 6A). Taken together, HYAL6 has no hyaluronidase activity. As mentioned above, sperm HYAL owns a ZP-binding domain at the C-terminal. If HYAL6 ZP-binding domain is involved in binding to eggs, the fertilization rate through IVF will decrease. Figure 6C shows that the fertilization rate was normal when IVF was performed after removing the COC layer surrounding the egg. Thus, we confirmed that the HYAL6 ZP-binding domain has no direct relationship with the binding to ZP.

## 5. Conclusions

This study confirmed that sperm HYAL function is essential for mammalian fertilization. However, HYAL6 did not have HYAL activity and was not directly involved in ZP binding in mice. Evolutionarily, HYAL6 is an extinction gene, but research regarding novel HYAL-like proteins is still beneficial as these proteins may inhibit fertilization among different species.

## Figures and Tables

**Figure 1 genes-13-00753-f001:**
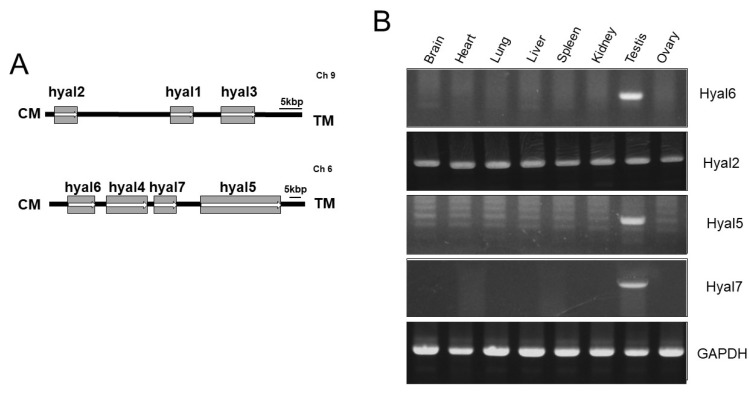
Chromosomal orientation and expression patterns of mouse hyaluronidase (*HYAL*) genes. (**A**) Seven genes (*HYAL1* through *HYAL7*) are located on chromosomes six and nine. Arrows (→) indicate the protein-coding direction. TM, telomere; CM, centromere. (**B**) Reverse transcription-polymerase chain reaction (RT-PCR) analysis. Mouse tissue complementary DNA was amplified by PCR, and the fragments were then stained with ethidium bromide. GAPDH; glycer-aldehyde-6-phosphate dehydrogenase.

**Figure 2 genes-13-00753-f002:**
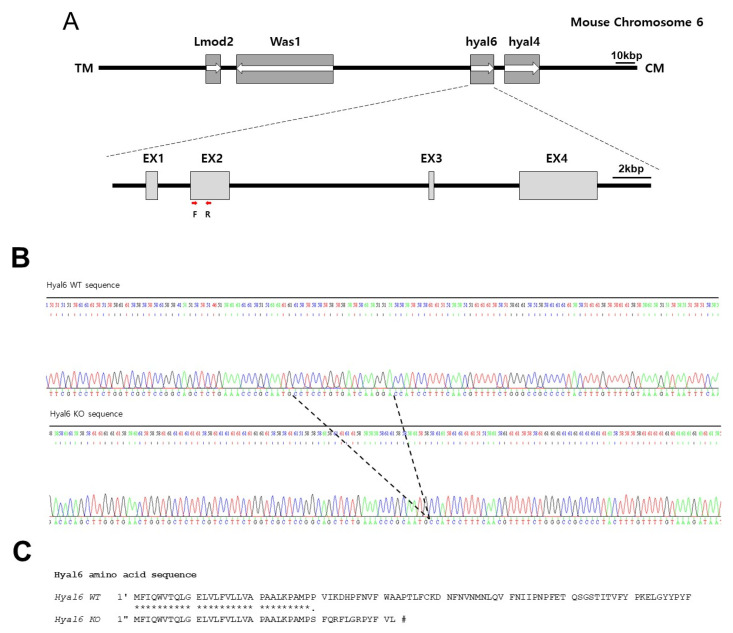
Generating hyaluronidase 6 (*HYAL6*) knockout mice using clustered regularly interspaced short palindromic repeats (CRISPR)/CRISPR-associated protein 9. (**A**) Mouse *HYAL6* contains four exons interrupted by three introns. The ATG translation initiation codon and stop codon are in exons two and four, respectively. F and R indicate primers for genomic PCR analysis. (**B**) Genomic sequence of the CRISPR target site in *HYAL6*. Letters inside the dotted lines indicate the target deletion. (**C**) Asterisks (*) indicate amino acids shared between the two sequences, and the hashtag symbol (#) indicates the stop codon.

**Figure 3 genes-13-00753-f003:**
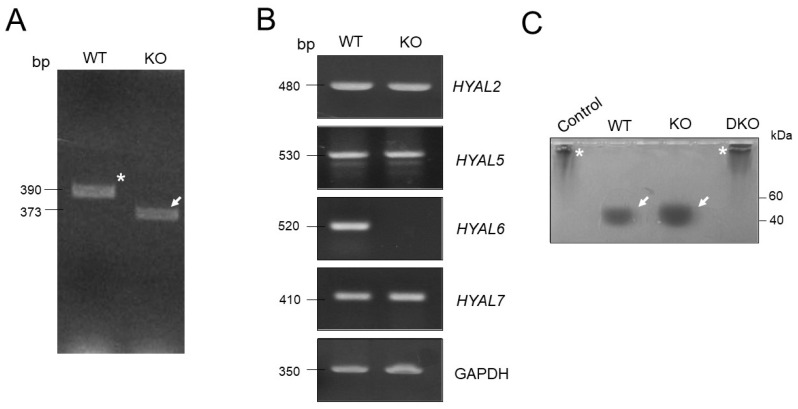
Characterizing male HYAL6 knockout (KO) mice. (**A**) Genomic DNA polymerase chain reaction (PCR) results from wild-type (WT) and KO mice. An asterisk (*) indicates a WT band, and arrows (→) indicate Hyal6 KO. (**B**) Reverse transcription PCR analysis of *HYAL2*, *HYAL5*, *HYAL6*, *HYAL7*, and glycerol-3-phosphate dehydrogenase (i.e., *GAPDH*) using complementary DNA from WT and KO tissues. (**C**) Sperm-specific hyaluronan-hydrolyzing hyaluronidase activity test. Two asterisks indicate high-molecular-weight hyaluronic acid, while two arrows (→) indicate low-molecular-weight hyaluronic acid digested by sperm hyaluronidase.

**Figure 4 genes-13-00753-f004:**
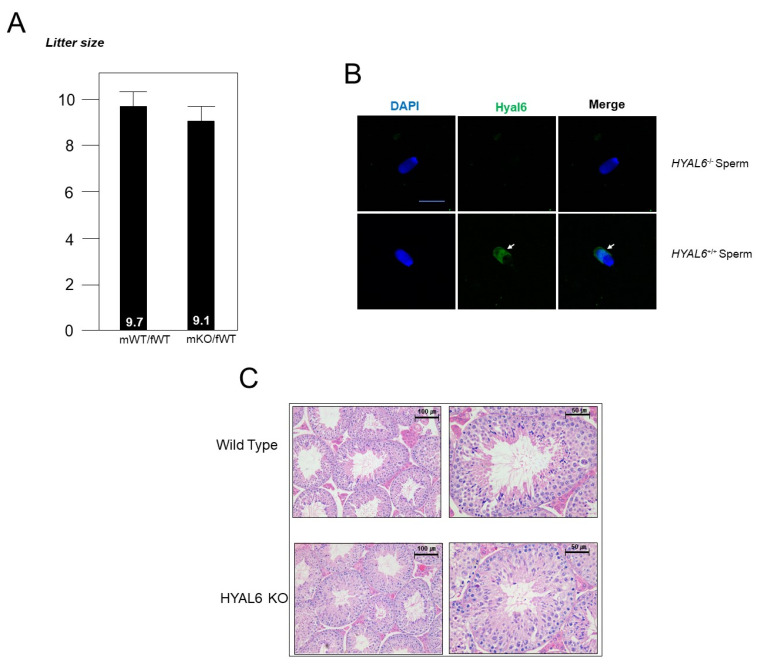
HYAL6 KO does not affect spermatogenesis or the fertilization rate. (**A**) Mean litter size from HYAL6 KO male mice. mWT, male wild-type; fWT, female wild-type; mKO, male knock-out. (**B**) Indirect immunofluorescent analysis of capacitated epididymal sperm. The HYAL6 molecule was probed by affinity-purified anti-HYAL6 antibody in WT sperm. No significant staining was observed in the KO sperm. Sperm were also stained with 4′-6-diamidino-2-phenylindole (i.e., DAPI). The arrows (→) indicate mouse HYAL6 protein expression in WT sperm. There was no staining in HYAL6 KO sperm. (**C**) Hematoxylin and eosin staining of the HYAL6 KO mouse testis. Losing HYAL6 does not affect spermatogenesis or the fertilization rate.

**Figure 5 genes-13-00753-f005:**
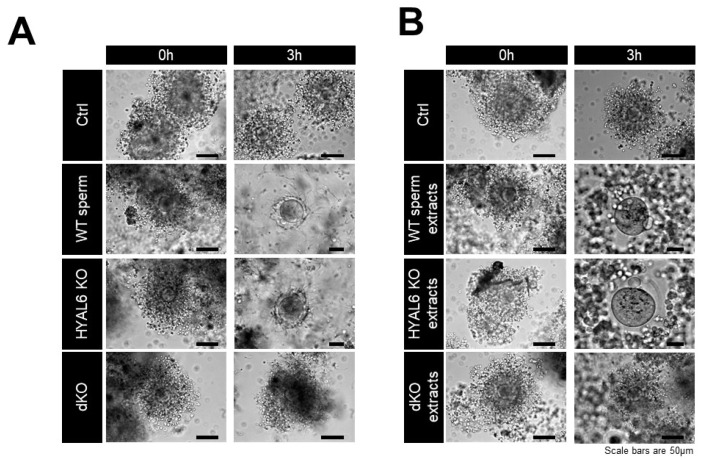
Cumulus cell dispersal by HYAL6 KO sperm and sperm extracts. A. The cumulus–oocyte complexes (COCs) were incubated with KO sperm (**A**) and KO sperm extracts (**B**).

**Figure 6 genes-13-00753-f006:**
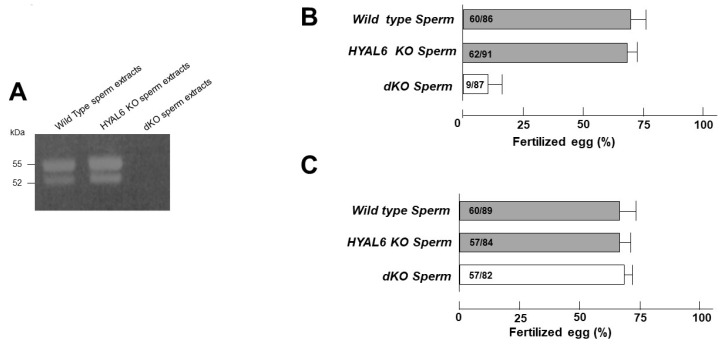
Zymography and in vitro fertilization assay. (**A**) Sperm protein fractions were prepared from wild-type mice, HYAL6 KO sperm, and HYAL5/HYAL7 double-KO (dKO) sperm; then, 30 µg of the sperm extracts were separated by 0.1% hyaluronan sodium dodecyl sulfate-polyacrylamide gel electrophoresis under non-reducing conditions and analyzed by zymography. (**B**) Fertilization ratio of intact eggs inseminated with capacitated cauda epididymal sperm from HYAL6 and dKO (open column). (**C**) Fertilization ratio was determined after inseminating cumulus-free eggs with capacitated cauda epididymal sperm derived from HYAL6 KO (shaded column) and dKO (open column) mice.

## Data Availability

Not applicable.

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
