# Peer review of "Hyaluronidase 6 Does Not Affect Cumulus–Oocyte Complex Dispersal and Male Mice Fertility"

_genes, 2022, doi:10.3390/genes13050753_

Round 1

Reviewer 1 Report

This paper has demonstrated that Hyal6 is not essential for fertilization. Data are appropriately obtained and the conclusion seems to be correct. However, it is not clear whether Hyal6 protein itself has an enzymatic activity of hydrolase or not. The enzymatic activity of the recombinant Hyal6 protein should be measured. If it has the activity, the cellular localization and stability of the Hyal6 protein should be investigated. This is a very important issue, the information of the enzymatic activity of the Hyal6 protein should be added.

Majot point

  1. Measure the enzyme activity of the Hyal6 protein.
  2. If it has the enzyme activity, the reason why it is dispensable for fertilization should be investigated. (Ex. Its cellular localization, stability and so on)

Minor points.

  1. References are not cited in the text at all.
  2. Line 235. Figure3C -> Figure 3C
  3. Line 300. “could not” Is this correct? Isn’t it “could”?

Author Response

We would like to thank you for possible accepting our manuscript to be published in genes. We studied each comment carefully to incorporate it in our final manuscript. We are certain that the reviewer’s comments were very helpful in improving the quality of our manuscript, and we greatly appreciate the reviewer’s effort.

Eminent scholars in reproduction and development participated in this study. The answer to each comment is shown below. Thank you again for your effort.

  1. Measure the enzyme activity of the Hyal6 protein.

Answer: As the reviewer mentioned, we also tried to measure the activity of Hyal6 protein. After cloning the HYAL6 gene in pCXN2, a mammalian expression vector, transfection was performed on HEK293 cells. Although we confirmed hyaluronidase activity, but unfortunately, there no hyaluronidase activity in transfected HEK293 cells. Figure 6A shows that the Zymography experiment using WT and hyal6 KO sperm extracts showed no difference in hyaluronidase activity in WT and KO.

  1. If it has the enzyme activity, the reason why it is dispensable for fertilization should be investigated. (Ex. Its cellular localization, stability and so on)

Answer: Based on our experimental results, we concluded that there was no hyal6 hyaluronidase activity.

Minor points.

  1. References are not cited in the text at all.

Answer: We checked the reference and text and re-arranged them accordingly.

  1. Line 235. Figure3C -> Figure 3C

Answer: As the reviewer mentioned, we re-wrote it.

  1. Line 300. “could not” Is this correct? Isn’t it “could”?

Answer: Yes. Hyal6 KO sperm extract was able to decompose COC. We revised the sentence.

Reviewer 2 Report

The present study entitled- Hyaluronidase 6 does not affect cumulus-oocyte complex dis-2 persal and male mice fertility by Bang and Colleagues showed the importance of Hyaluronidase 6 in male reproduction. Authors have done a very deep and thorough study by different biological methods. This study sound good and executed nicely as well. However, after a careful review of this manuscript the following needs to be addressed during the revision.

Major

Q1. The listed references have not been written in the text body. Which is a very serious issue?

Q2. A rationale of the study should be clearly written in introduction.

Q3. Zymographic assays for Hyal enzyme activity confirmed no significant difference in wild-type and HYAL6 KO sperm extracts. Author did explain this discrepancy?  Can author explain in the discussion.

Q4. Thus, based on these findings, we concluded that HYAL6 is not essential for mouse spermatogenesis. Based on histological finding only it is quite difficult to conclude, there is no role in spermatogenesis. As author used HYAL6 KO mice (probably in testis), testis performs spermatogenesis and steroidogenesis, based on this some queries needs to be clarified.

Does testis express HYAL6?

It would be important to show the localization of HYAL6 in testis control and in the KO mice which particular cell type has been knock out  for HYAL6.

Minor

Q1.Line 40- Genetic engineering- add genetic

Q2.Line42-43- Rodection time- What is rodection time??

Q3.Line43-44- However, some genes are not critical for cell survival. Which gene does author mean?? Be precise with some examples.

Q4.Line47-48- This sentence is very general. Did author focus mammal or other vertebrates??

Q5.Zymography assay- What the substrate for zymogen assay?

Q6. Sample size is not mentioned.

Author Response

Thank you very much for your kind review. In fact, what the reviewer mentioned was very helpful in improving the quality and understanding of our manuscript. Major points and Minor points pointed out by the reviewer were faithfully reviewed. We want reviewer to check our answers and take a good look at them.

Major

Q1. The listed references have not been written in the text body. Which is a very serious issue?

Answer: We checked the reference and text and re-arranged them accordingly.

Q2. A rationale of the study should be clearly written in introduction.

Answer: As the reviewer suggested, we have supplemented the introduction part.

Q3. Zymographic assays for Hyal enzyme activity confirmed no significant difference in wild-type and HYAL6 KO sperm extracts. Author did explain this discrepancy?  Can author explain in the discussion.

Answer: As the reviewer pointed out, we seem to lack analysis of the active results for hyaluronidase. It was described in more detail in the discussion part.

Q4. Thus, based on these findings, we concluded that HYAL6 is not essential for mouse spermatogenesis. Based on histological finding only it is quite difficult to conclude, there is no role in spermatogenesis. As author used HYAL6 KO mice (probably in testis), testis performs spermatogenesis and steroidogenesis, based on this some queries needs to be clarified.

Does testis express HYAL6?

Answer: Yes. HYAL6 is expressed in the testis. See Fig. 1b. And, we considered that hyal6 had nothing to do with sperm formation, considering that HE staining did not show much difference in hyal6 KO and WT.

It would be important to show the localization of HYAL6 in testis control and in the KO mice which particular cell type has been knock out for HYAL6.

Answer: Figure 4C is an experiment comparing WT and KO testis using HE staining. A normal spermatogenesis could be confirmed at both sites. Therefore, we thought that normal spermatogenesis also occurs in HYAL6 KO testis. In fact, spermatogenesis was normally performed when the sperm-specific HYAL5 and HYAL7 were knock out in mouse. In other words, it can be implied that sperm-specific hyaluronidase is also found in testis, but is not involved in spermatogenesis.

Minor

Q1. Line 40- Genetic engineering- add genetic

Answer: As mentioned reviewer, we mis-wrote. We added genetic.

Q2. Line42-43- Rodection time- What is rodection time??

Answer: We are miss-written. We added p.

Q3. Line43-44- However, some genes are not critical for cell survival. Which gene does author mean?? Be precise with some examples.

Answer: As the reviewer pointed out, we added more explanation. See introduction

Q4. Line47-48- This sentence is very general. Did author focus mammal or other vertebrates??

Answer: Yes. Reflecting the reviewer's opinion, we revised the sentence a little.

Q5. Zymography assay- What the substrate for zymogen assay?

Answer: We used high molecule Hyaluronan in the zymography assay

Q6. Sample size is not mentioned.

Answer: As the reviewer mentioned, we added sample size. See Fig 3C.

Round 2

Reviewer 1 Report

This paper has demonstrated that Hyal6 is not essential for fertilization. Data are appropriately obtained and the conclusion seems to be correct. However, the authors do not mention that Hyal6 protein itself has no enzymatic activity in the text. Although it is negative data, this point is very important. The methods and results of the preparation of recombinant Hyal6 as well as its measurement of the enzyme activity should be described. If it has already been reported, the reference should be cited. I also think a possibility that Hyal6 may act only on chondroitin sulfate but not on HA, such as Hyal4. Have the authors measured the activity of Hyal6 toward chondroitin sulfate?

Major point: It should be described that Hyal6 protein itself has no enzymatic activity in the text.

Minor points:

  1. Line numbers 68, 103, 364, and 378.

Delete hyphens; HY-AL6, tar-get, There-fore, and con-firmed.

  1. Line numbers 131, 140, 147, 147, 155, 172, 179, 187, 189, 192, 198, 203, and 206.

Delete space between value and %.

  1. Line number 113. “approximately 373-bp”

Delete “approximately”. Or “approximately 370 (400?)-bp”

  1. Line number 189. CO2. 2 should be a subscript.
  2. Line 373. hyaluronidase -> HYAL

Author Response

Dear reviewer

Thank you very much for your kind review. In fact, what the reviewer mentioned was very helpful in improving the quality and understanding of our manuscript. Major points and Minor points pointed out by the reviewer were faithfully reviewed. We want reviewer to check our answers and take a good look at them.

This paper has demonstrated that Hyal6 is not essential for fertilization. Data are appropriately obtained and the conclusion seems to be correct. However, the authors do not mention that Hyal6 protein itself has no enzymatic activity in the text. Although it is negative data, this point is very important. The methods and results of the preparation of recombinant Hyal6 as well as its measurement of the enzyme activity should be described. If it has already been reported, the reference should be cited. I also think a possibility that Hyal6 may act only on chondroitin sulfate but not on HA, such as Hyal4. Have the authors measured the activity of Hyal6 toward chondroitin sulfate?

Answer: We think the reviewer's point is completely correct. When HYAL6 was cloned to a mammalian expression vector (pCXN2 vector) and confirmed for hyaluronic acid (HA) activity, there was no HA decomposition activity. However, we have not checked whether chondroitin sulfate can be decomposed by HYAL4. In the end, since it was important whether cumulus-oocyte-complex (COC) could be disperse in this study, HYAL6 only examined its function in mammalian fertilization.

As the reviewer pointed out, whether Hyaluronidase can disperse chondroitin sulfate seems to be an interesting topic. In the next paper, we will discuss these things.

Major point: It should be described that Hyal6 protein itself has no enzymatic activity in the text.

Answer: As a result of hyal6 KO mouse function analysis, we described that HYAL6 don't has enzyme activity at the end of discussion part. There was no hyaluronidase activity in HYAL5/HYAL7 male KO sperm extracts, and HYAL6 KO sperm extracts clearly showed HYAL5 and HYAL7 hyaluronidase active bands. As mentioned above, HYAL6 seems clear that hyaluronidase activity is absent. As the reviewer mentioned, we would like to examine whether HYAL6 is active other than hyaluronidase activity in the next paper.

Minor points:

Line numbers 68, 103, 364, and 378.

Delete hyphens; HY-AL6, tar-get, There-fore, and con-firmed.

Answer: We are terribly sorry. There was a mistake in our process of changing formets. I changed it to what the reviewer pointed out.

Line numbers 131, 140, 147, 147, 155, 172, 179, 187, 189, 192, 198, 203, and 206.

Delete space between value and %.

Answer: There is no space between the value and % in everything we have written.

Line number 113. “approximately 373-bp”

Delete “approximately”. Or “approximately 370 (400?)-bp”

Answer: We changed 373 to 370.

Line number 189. CO2. 2 should be a subscript.

Answer: We changed.

Line 373. hyaluronidase -> HYAL

Answer: We changed.